# A Review of Research on Tree Risk Assessment Methods

**Haibin Li** [1,2], **Xiaowei Zhang** [1,2], **Zeqing Li** [1,2], **Jian Wen** [1,2,*] and **Xu Tan** [1,2]

1 School of Technology, Beijing Forestry University, Beijing 100083, China
2 Joint International Research Institute of Wood Nondestructive Testing and Evaluation, Beijing Forestry University, Beijing 100083, China
* Correspondence: wenjian@bjfu.edu.cn; Tel.: +86-158-0146-7078

**Abstract:** As an important part of the urban environment, trees have certain risks while living in harmony with humans. For example, the failure of trees in extreme weather may cause casualties and damage to public and private; the decline and death of old and valuable trees can have an impact on the diversity and cultural value of trees. This paper outlines the theories related to tree risk and the development of tree risk assessment, evaluates the advantages and disadvantages of various tree risk assessment methods in existing studies, and explains some factors affecting the bearing capacity and related applications using knowledge of tree mechanics. Approaches in modern probing techniques are applied to study the response and loading of tree crowns and branches under wind loads, the application of different non-destructive testing techniques in visual assessment for detecting internal defects and root distribution of trees, and the role and impact of objective quantitative test results on tree risk assessment. Finally, the future development direction of tree risk assessment is predicted, which provides an important reference for research on tree risk assessment.

**Keywords:** urban tree; tree risk assessment; non-destructive testing; risk assessment methods

## 1. Introduction

With the increasing growth of the urban economy, the destruction of the urban ecological environment is becoming more serious. Urban construction concepts such as "garden city", "forest city", and "green city" are used to achieve a balance between ecology and economy and to prioritize ecology in development [1]. Urban trees are the top priority in the improvement of urban ecological enhancement [2]. However, the ecological environment problems brought by urban development will cause trees to suffer from various "urban diseases" [3]. As trees grow, there is the possibility of branches dying or rotting at some point, or loss of strength due to mechanical damage caused by exposed wounds or decay caused by pests and diseases, causing the trees to fail over in extreme weather, even though they bring many positive values to the people who live and work in urban environments [4]. There is no doubt that tree failure has affected the safety of people and property as well as the traffic in urban areas.

All trees have the potential to pose a certain degree of risk to nearby people, buildings, and public facilities. However, the likelihood of a safety incident tends to be minute and far less than trees' ecological, social, and economic benefits. In the case of older trees, conditions such as aging, pests, and diseases would cause the trees themselves to weaken (such as the decay of the trunk, root damage, and trunk tilt). At that point, the tree structure is no longer reliable, and the tree can constitute a threat to people and surrounding buildings [5], which is particularly evident in old trees. Urban trees, as distinct from forest trees, are characterized by complex growth environments, greater vulnerability to damage, and wind tunnel effects caused by high-rise construction [6]. Appropriate preventive and protective measures should be taken to effectively avoid the occurrence of hazards, if tree hazards can be detected in advance.

Tree risk assessment is a systematic process of identifying, analyzing, and determining tree risks to detect risks before they cause safety incidents and determining the level of risk

concerning the location, extent, and possible impact on the surrounding environment [4]. After determining the risk level of a tree, tree risk management is needed to reduce risk to protect people and property based on the rank. Risk mitigation measures can be specific to improvement of tree growth conditions, aiming to reduce the number of safety incidents that occur with trees, or can be based on impact goals, to reduce the consequences of impacts that occur when a safety incident occurs [7].

In addition to risk reduction, risk management determines the frequency of inspection of assessment targets, which is determined by assessing the impact target and area of the target tree. It is imperative to assess tree risk as often as necessary in areas with frequent human activity and vulnerable target areas. Moreover, the need for tree risk assessment is low in areas off the beaten path or where no buildings exist [8]. This review provides recommendations for future research aimed at improving the validity of tree risk assessment methods.

## 2. Concepts Related to Tree Risk Assessment

### 2.1. Risk vs. Hazard

In the early days of assessment practices, arborists and forestry staff used "hazard assessment" to describe the specifics of examining and assessing the structural condition of trees and the potential hazards associated with tree safety issues [4,9]. Risk assessment is the standard term now adopted since the publication of Pokorny's book Urban Tree Risk Management: A Community Guide to Program Design and Implementation. [10]. Andrew K. Koeser (2016) [11] pointed out that the difference between risk and hazard in determining the likelihood of harm is that if something is only likely to cause harm, no matter how big or small, it should be regarded as a hazard. In contrast, risk refers to the possibility of harm from a potential hazard, and the magnitude of risk depends on the actual situation. In addition, risk is the combination of the likelihood of an event and the severity of the potential consequences [12]. The most comprehensive and authoritative definition of risk is from the Tree Risk Assessment Manual, published by the International Society for Arboriculture. Ryan W. Klein [13] defined risk as to the likelihood of an event occurring and the severity of its potential consequences. There are three elements to consider in determining the risk of a tree: (1) the ability to cause a hazardous accident or consequence, (2) the likelihood of an accident occurring, and (3) the specific target.

### 2.2. Risk Target

This term denotes people, property, or activities injured, damaged, or disrupted by a falling tree [14]. In comparison, Norris [15] mentioned that risk targets can more accurately represent the concept which is affected by the falling tree and avoid the term target, which may be taken to suggest what to aim for.

### 2.3. Risk Assessment and Health Evaluation

Tree health describes the growth condition of trees and focuses on protecting and restoring trees to a healthy state, while risk assessment concentrates on mitigating failure caused by trees. When a tree has good growing status and a reasonable structure, it can tolerate adverse environments, diseases, and pests. In contrast, unhealthy trees can break, fall, and wither, putting pedestrians and surrounding buildings and facilities in danger. This ability, known as mechanosensitive control, is an adaptive feature that plants have in response to changes in growth and morphology due to environmental changes and can reduce the risk of breaks. Therefore, one of the purposes of determining the health status of a tree is to anticipate potential risks to the tree, and the most important basis for determining tree risk assessment is the health status of the tree.

Risk assessment and health evaluation are related but not identical, and the emphasis of each study is different. Health evaluation focuses on observing the state and shape of the tree crown, branches, roots, etc. In contrast, risk assessment emphasizes the structure

and internal condition of the branches and trunk as well as the distribution of roots, and needs to consider the risk targets of the tree [16].

## 3. The Development of Tree Risk Assessment Studies

A core demand for tree risk assessment research is tree security, and the critical factor promoting the development of this research in the United States is the liability disputes faced by tree management departments in lawsuits for injuries or property damage caused by safety incidents with trees [17] and the costly maintenance problems associated with tree damage [18]. Consequently, professional tree management organizations, the International Society of Arboriculture (ISA), and various professional arborist certifications have been established. In contrast, the need for greening in China, mainly from landscape management by the park Bureau responsible for preservation of old trees, and the increasing number of tree safety incidents have driven the development of tree risk assessment [19].

The development of tree risk assessment could be divided into systematic risk assessment developed by summarizing the characteristics of failed trees at the accident site, assessment methods that apply knowledge of tree mechanics to interpret tree carrying capacity as a partial reference factor, and probing tools and detection methods developed based on needs when visual inspection does not yield a comprehensive risk assessment. These three types of approaches are more than a simple timeline development, they are different levels of assessment of different risk levels and they can be combined into a complete tree risk assessment and risk management.

### 3.1. Systematic Tree Risk Assessment

The research on tree risk assessment goes back to the 1960s–1970s when Wagener, a plant pathologist, summarized common tree defects in recreational sites and frequent hazards of significant tree species based on his work experience [9,17], the FRST manual. Paine collected a large amount of tree failure and accident data (species, failure category, target, damage, etc.) in the form of data cards, stored on data tapes for research retrieval [20,21], on the basis of which he proposed to rate the hazard level as the product of the probability of failure, probability of impact, the damage potential of possible failure, and the target value [22]. The applicability of the system is limited by the subjective nature of the last three estimates. In addition, considerable expertise is required to make accurate estimates, and separate assessments are time-consuming. The applicability of the system was limited to forest recreation sites and specific trees included in the database, and although later modifications and extensions were made on this basis, such shortcomings could not be avoided [23]. G.W. Hickman [23] proposed an evaluation system of eleven assessment indicators based on four aspects of the tree's environment, structure, vigor, and target, each with a scale of 1 to 5. A standard field evaluation form was developed to assess 1400 oak trees, and the evaluation process offered a relatively objective and reliable analysis of the condition of trees that may affect their hazard potential. Then, after seven years, the system was applied again to evaluate and compare the current status of these oaks [24], and three indicators were selected that could correctly reflect the weakening of the trees in terms of tree vigor, trunk condition and degree of tilt. The equation Y = $-8.876 + 1.696 \times$ vigor $+ 1.696 \times$ trunk $+ 1.696 \times$ tilt, where Y leads to a prediction of tree failure (Y > 0) or survival (Y < 0), is used, and it was pointed out that the tree strength was judged by leaf color and degree of cover. The ISA Tree Hazard Evaluation Method, which combines failure potential, size of possible failure site, and target rating, with the three scored on a scale of values 1 to 4 and the sum of the three reflecting the overall hazard rating, has been extensively applied and modified by municipalities and commercial arborists [11,13]. However, because only the sum of the three ways to determine risk grade cannot fully reflect the risk of trees, for example in the own existence of very big security hidden dangers but no measures of trees as target level of risk, it is concluded that the tree are at high risk, but in fact, since there is no risk of target, the tree does not cause the risk [19].

For this type of tree risk assessment, which consists of several qualitative values, the numbers are assigned to considerations based on the severity of the risk to arrive at an estimate or ranking of relative risk. The advantage of numbers over textual descriptions is that they are unambiguous and easy to interpret, making it possible to determine risk visually. In order to improve the reliability and consistency of the results, the terms and significance of likelihood, consequence, and risk ratings require explicit definition [12]. The numbers conferred by the results categorize rather than quantify risk. The final method of arriving at the relative risk levels for which the trees provide a reference is adding or multiplying these numbers together. However, this treatment of the numbers can magnify certain tolerable factors, such as an endangered tree growing in an off-the-beaten-path area with low risk but a high calculated score. The risk rating matrix was introduced in ISA's latest BMP Method (Best Management Practices for Tree Risk Assessment): The probability of failure (never possible, possible, likely to happen soon) and the probability of influencing the target (from very low, low, medium to high) are separately assessed and incorporated in the risk matrix to assess the likelihood that a tree will fall and impact the target [13]. There are, however, a number of drawbacks to risk matrices, and Cov (2008) [25] found that it is possible to assign more considerable qualitative risks to quantitatively more minor risks (reverse ranking), resulting in qualitative health risk ratings that are contrary to reality. Moreover, the BMP method satisfies Cov's requirements for risk matrix validity. In 2005, Ellison [8] introduced the concept of probabilistic quantification of tree risk, which is different from the previous qualitative assessment. In the process of assessment, a quantitative tree risk assessment (QTRA) system is developed by using quantitative processing of the obtained information and probabilistic description of risks. The system is an expansion of the concepts proposed by Paine (1971), Helliwell (1990, 1991), and Matheny and Clark (1994), and follows the definition of tree hazards proposed by Matheny and Clark and renames it: (1) probability of failure, (2) size, and (3) target, and the product of the three is the risk of harm (ROH), which assesses the likelihood of assessing a target site to be occupied by vehicles or pedestrians through the calculation of vehicle time occupancy on the road and dwell rate within the tree safety zone, classifying potential impacts based on the tree trunk and branch diameters. By quantifying the independent probabilities of these three components, the QTRA system calculates the resulting risk of injury compared to an acceptable level of risk, with a probability of death or severe injury of 1/10,000 as the limit of acceptable risk to the public [8], and all risks with probabilities above 1/10,000 will be remediated to reduce the risk to an acceptable level. This quantitative assessment is able to quantify the risk of injury from tree failure and its use of quantitative risk assessment (QTA) is based on reliable decision support principles [26], but as trees are natural structures, the degree of probabilistic quantification is limited, and the assessment of the probability of the hazard occurring in trees relies more on subjective assessments [15]. While this approach is called quantitative, it is worth pointing out that there is no authentic quantitative approach. All risk assessments, to some extent, need personal judgment, especially for the probability of failure [13].

In order to calculate the ultimate risk level more accurately, Weng Shifei, Li Cai Min, et al. [27] constructed a tree health evaluation system by selecting 14 intuitively integrated indicators from four aspects: overall condition, crown, trunk, and roots of trees based on a greenfield survey combined with expert opinions. The relative weights of each index were determined by hierarchical analysis (AHP), a combined qualitative and quantitative decision analysis method for solving complex problems with multiple objectives. The study by Chen Junqi [28] evaluated the health of ancient trees in Beijing by analyzing morphological and ecological characteristics, selecting indicators with apparent features, evaluating morphological indicators with hierarchical analyses, and analyzing ecological indicators with principal components analysis and gray cluster analysis. Finally, the two analysis methods were compared using BP neural network to obtain reliable evaluation results. Such evaluation approaches that combine decision-making and statistics provide a new area of research for quantifying tree risk assessment.

Although there are some significant differences in these assessment methods, all systematic assessment methods include tree structural evaluation, defect identification, assessment of the probability of failure of the tree, assessment of the risk target, and assessment of the damage caused by impact on the risk target [4,8,29–32]. In addition to these similarities, assessment methods have different approaches on assessing different defects, integrating various potential risk factors, and combining various components into a final and comprehensive risk assessment result [15,31]. The applicable conditions of each method should be considered as the basis for risk assessment. It is worth noting that, in contrast to the methods used to assess urban trees, the assessment of forest trees [33] takes into account more forest-based characteristics such as species composition, degree of stand density, forest type, etc., which are not considered in this review.

### 3.2. Risk Assessment Based on Tree Mechanics

Systematic tree risk assessment methods that rely on empirical observations and statistics can analyze qualitatively or, to some extent, quantitatively, the risks to trees but cannot explain why trees are at risk. Research from a tree mechanics perspective can address uncertainties in the perception of tree hazards. If a systematic risk assessment originates from a field survey of tree failure, an assessment combined with knowledge of tree mechanics is a series of judgment methods based on defects that affect the tree's carrying capacity. Representative methods include the visual tree assessment (VTA) method [29], which focuses on the external manifestations of internal defects; the static integrated analysis (SIA) method [34], which takes into account factors such as tree height, crown shape, and wood strength to assess the rupture strength of hollow trees; the Integrierte Baumkontrolle (IBA, from Germany) method [35], which describes the interaction of mycology, vitality, and stability to identify decay before tree failure; and the Wessolly method [36], etc. Therefore, an increasing number of people are using tree mechanics principles to understand the carrying capacity of urban trees, to interpret and propose options for a subset of options, and to more fully develop risk assessment guidelines and techniques to measure the potential for tree hazards. Studies on tree mechanics have focused on wood properties, structure, tree defects (mainly trunk), and root distribution of trees.

Material properties of trees are measured in many tests, but their variability is considerable, and measurements differ depending on the age, growing conditions, species, moisture content, and location of the tree [37–46]. The properties most commonly used to reflect the properties of tree materials are the modulus of elasticity (E) and the modulus of rupture (MOR) [47]. Although many material properties of wood have been described and mean values recorded [40], it is often the variability of wood material that leads to the low availability of these values on live wood [48]. This is due to the fact that E and MOR decrease with increasing trunk height and branch length [41,49], and E values decrease as the tree matures. Bouslimi (2014) [50] compared samples of wood from healthy and decayed sections of eastern white cedar (*Thuja occidentalis* L.). The relationship between decay on mechanical properties and weight loss was significant, with a 40% loss of MOR and 30% loss of E resulting from a 15% weight loss.

The influence of the tree's structure on the tree's carrying capacity lies mainly in the length and diameter of the trunk and branches and the direction of the load. The length of the trunk and branches affects the bending moment and torque generated by the load. When subjected to the same load, the longer the branch and trunk are, the more torque they will be subjected to [47]. By the material mechanics knowledge, the carrying capacity of the tree trunks and branches to its cross-sectional area and moment of inertia (hereinafter referred to as "I").

$$\sigma = \frac{M_e y}{I_z} \tag{1}$$

$$I_z = \frac{\pi D^4}{64} \tag{2}$$

From Equation (1), it can be seen that the positive stresses acting on the cross-sections of the trunk and branches are related to the cross-sectional bending moment $M_e$ and the area moment $I_z$. It can be seen from Equation (2), the area moment of the circular section, that the effect of diameter on the bearing capacity of trunks and branches is nonlinear: the cross-sectional area is proportional to the square of the diameter, and $I_z$ is proportional to the fourth power of the diameter. As such, the heartwood and sapwood contribute disproportionately to the bearing capacity, and studies [51] indicate that sapwood provides most of the mechanical support required by the tree. Researchers have used slenderness [52], defined as the ratio of length to diameter, as one of the indicators of tree stability and an indicator of the tree's ability to resist wind and snow damage.

Bruchwald (2010) [53] considered the inverse of the slenderness ratio as an essential consideration in his study of the risk assessment of wind damage to trees. Tsutomu (2012) [54] studied the influencing factors of trees' resistance to uprooting ability, and selected three artificial spruce forests with different characteristics, A, B, and C, to compare the relationship between the slender ratio and the ratio of the critical wind speed and the dead load to the critical moment of uprooting in the pulling test. The results found that trees with small slenderness ratios resist uprooting at higher wind speeds, and the ratio of self-weight load to critical moment increases gradually with the slenderness ratio. Slater (2015) [55] found that the bark inclusion significantly affects the strength and is related to the degree of bark blockage at the bifurcation through rupture tests on hazel trees. In 2020, Slater [56] performed tensile experiments on the bifurcation of the inclusion bark in order to explain the phenomenon of bulging at the intersection of the inclusion bark. The experimental results show that the bulge is a compensatory growth for the lack of strength and three important factors that influence the maximum bending moment of these branch connections: the morphology of the containing bark, the diameter ratio of the branch connection, and the width of the containing bark at the apex of the connection. Kontogianni (2011) [57] assessed tree stability based on above-ground silvicultural characteristics and selected canopy aspect ratio, canopy asymmetry index, and tree height as the most significant indicators of tree stability. In 2020, Kong [58] compared 13 formulas for calculating the canopy asymmetry index and selected the parameter that best reflects the true canopy shape.

Tree defects can lead to tree hazards such as hollows, cracks in bark and branches, cracking of the trunk, and decay [59], of which decay has been more studied. Decay is a natural process of fungal decomposition of wood [60,61], and the loss of wood reduces the carrying capacity of the tree trunk or branches. Therefore, biomechanical studies related to tree decay are critical in tree risk assessment. The current common method of strength calculation is to model the tree as a cantilever beam. In the calculation of bending stress, the moment of inertia $I_z$ of the cross-section is required, which increases exponentially with the increase of the trunk diameter. The presence of decay causes a decrease in $I_z$, the extent of which depends on the size and location of the decayed area. There is an exponential decrease in the $I_z$ value for larger regions compared to smaller regions. When off-center decay occurs in the cross-section, $I_z$ decreases exponentially even if the decayed area is the same. Studies assessing decay damage to trees began with Wagner's observations on conifers growing in the Pacific Northwest of the United States and found that collapse was more likely to occur when trunk decay (or hollowing) was at 70% [62]. Although Wagner stated that this finding should not be directly applied to other tree species, the observation of statistically safe and damaged trees after the hurricane confirmed the validity of his findings [63]. Then, Wagner (1963) proposed an equation to assess the likelihood of trunk hazard using the trunk diameter ($d_o$) at the direct ($d_i$) decay of the decay (or hollow) cross-section as follows.

$$d_i^3/d_o^3 \tag{3}$$

A similar approach was proposed by Coder (1989) [64] as follows.

$$d_i^4/d_o^4 \tag{4}$$

These two equations show that the loss of a moment of inertia $I_Z$ is relatively small if the decayed area is located in the center of the trunk with a circular shape, and the accuracy of the equation decreases once the decayed area is not centered with the trunk section. Smiley and Fraedrich (1992) [63] modified Equation (2) to account for the cavity in the trunk as in Equation (5), where k is the ratio of the cavity opening to the corresponding trunk circumference. Their modification reasonably predicts the strength loss due to offset decay [65]

$$\frac{d_i^3 + k(d_o^3 - d_i^3) \times 100}{d_o^3} \tag{5}$$

It is important to note that the three formulas of Wagner, Coder, Smiley, and Fraedrich estimate the percentage loss of I by considering only the area of decay as a percentage of the trunk diameter at the site of occurrence as a way to assess the probability of the tree being at risk.

The sapwood gives the tree its main structural strength, with the heartwood contributing very little. An intact sapwood ring is an important factor in assessing the stability of a tree, so the thickness of the remaining wall is of wide interest, and when certain fungi attack the reaction zone in the sapwood, breaking through the tree's defense mechanisms, irregular geometries and sapwood rings that cannot be clearly defined are produced [66]. Mattheck (1993) [67] predicted the likelihood of trunk hazard based on the ratio of the intact wood thickness (t) to trunk radius (R), and the data statistically yielded a ratio greater than 0.3 for trunk hazard to occur. This equation also reasonably predicted offset decay [65].

$$t/R \tag{6}$$

However, Wessolly and Erb (1998) [68] came to the opposite conclusion, using a hollow beam-based approach to derive a lower thickness threshold. Sterken (2005) [69] conducted a comprehensive assessment of a 17.1 m tall eucalyptus tree and concluded that this threshold was out in the middle of the first two extremes.

Brian Kane [70] compared these four strength loss equations. There is a parabolic relationship between the strength loss and the hollow rate: the strength loss is slight until the hollow rate becomes large.

However, the calculation of $I_z$ is based entirely on geometry, and the above equations all assume that the trunk cross-section is circular, so there will be errors in the actual evaluation. Koizumi and Hirai (2006) [71] calculated the cross-sectional modulus of irregularly shaped decay sections using high-resolution images. In contrast, Ciftci (2014) [72] considered the strength loss in the irregular decay region. Burcham (2019) [73] collected acoustic tomography images of tree trunks for estimating the percentage reduction in cross-sectional modulus. Since the images are able to reflect geometric details such as irregular shape, offset of defects from the center, etc., they provide a more accurate estimate of the trunk bearing capacity than the simplified hypothesis. Reis (2022) [74] used ultrasonic tomographic images to improve the range of use of the equations, e.g., taking into account the irregular shape of the actual trunk cross-section and internal air rot, averaging the values of Equations (4) and (5) after measurement at multiple locations, and adding consideration of the eccentricity distance to Equation (3). This brings the equations closer to the real conditions of the tree trunk.

Root distribution is an important factor in the study of tree mechanics to support trees. The study of the tree root was performed by applying horizontal forces and observing the soil damage and uprooting processes in uprooted trees [75], as well as the modeling based on this, using numerical simulations to calculate the mechanical analysis of the tree root anchoring and being uprooted [76–78]. When horizontal forces are applied to the trunk, the soil moves downward on the leeward side and upward on the windward side. The root-plate underground on the windward side is the first to be damaged, and as the horizontal force increases, the damage gradually spreads to the ground on the windward side, and then the tree roots are uprooted [75]. The root-soil plate is the main root and growing soil in

the tapered area under the tree roots, similar to concrete reinforcement, which holds the tree in the ground. Root-plate plates play an important role in tree stabilization [79,80]. In the trenching tests of eucalyptus by Ghani et al., it was found that root damage firstly shadows torsional moments and that root depth is the main factor affecting the effectiveness of root anchorage [81]. However, in urban environments, compacted soil, underground facilities, environmental pollution, etc. may affect the growth and development of the root system. Dumroese et al. (2019) [82] studied trees on slopes and found that when trees are in a slope or prevailing wind geographic location, more roots grow downhill and windward to how the tree is stable, while Krisans (2020) [83] indicated that root decay can have a significant negative impact on tree stability, regardless of soil type. Therefore, probing the root distribution to understand the anchoring properties of tree roots can predict how trees will respond to weather such as storms [84].

The compression of trees by snow and ice is considered a static load, so wind-blown trees can be considered a load of shorter duration, i.e., a dynamic load [85]. Controlled pulling is an analysis method that reduces dynamics to statics, where wind actually causes trees to sway. In an early study of tree stability, Coutts (1986) [75] recognized that windthrow is a dynamic process. This is because wind loads are cyclical and open trees in cities have a large branch mass that will cause complex swaying of the trees. Wind tunnel tests [86–88] and tensile experiments simulating wind tunnels [89,90] have been used to study the effects of wind on trees, but are limited by the size of the wind tunnel and are only an approximate simulation of realistic results. Meanwhile, some complex models are built for tree dynamics analysis [91], mainly considering the contribution of branches to the dynamics [92], and geometric factors have a stronger influence on tree sway than material factors [93].

### 3.3. Visual Tree Risk Assessment

#### 3.3.1. VTA

Mattheck [29] proposed visual tree assessment (VTA) by combining biological and mechanical perspectives. This assessment considers the vitality of the tree, such as growth potential, old growth, branch sinking, and wound healing, as well as the degree of damage suffered by the tree and its stability against windfall, heavy rain, snow, etc. It is considered that the stress distribution is the same in the normal state of the tree before the damage. Mattheck's (1993) axiom of constant stress proposes that when the tree is damaged, it tries to return to a state of stress homogenization by proliferating at the damage site to compensate for the lack of support of the tree, producing a symptom of the repaired damage defect. For example, internal hollow rot can cause the surface of the trunk to bulge, and cracks can grow in the form of ribbed bumps (Figure 1). VTA considers these symptoms to be warning signs of the condition of the tree. If signs of defects have been noted, they must be confirmed with sophisticated testing and evaluation of the noted signs of defects. During extreme weather, such as typhoons, the canopy will gather wind like a sailboat and transmit the wind load through the trunk to the roots, creating intense bending stresses in the trunk or thick branches. When the wind load enters the root system, the wind load is distributed between the thick and thin roots, and the entire wind load is eventually carried by the soil surrounding the root system (anchoring effect).

The different stages and intensities of the VTA method assessment depend on the assumed severity of the suspected defect and consist of 3 steps.

- Appearance inspection to diagnose the growth state and structure of trees, investigate the scale of damage, decay, and cavity, and determine whether there are signs of danger in trees;
- Precision inspection: When a tree is found to have signs of danger, a diagnostic instrument is used to measure its internal decay, the presence or absence of cracks, and the strength of the tree's wood;
- Hazard determination, measurement and analysis of crucial defects, and calculation of residual strength of trees.

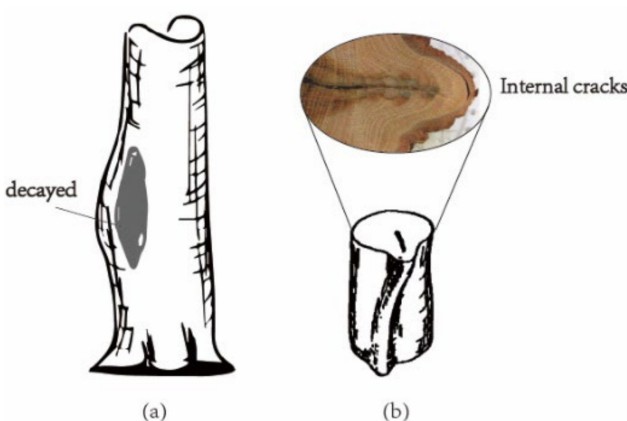

**Figure 1.** Damage and proliferation of trees, (**a**) bump caused by decay inside the trunk, (**b**) internal cracks forming rib ridges.

The diagnostic instruments are fractometer, sound impulse hammer, and increment borer. Among them, the fractometer is a mechanical measuring instrument for determining the characteristic values of wood bending and compression strength. It can provide a precise and quantitative determination of tree embrittlement, and the pulse hammer can detect defects such as decay, cracks, and closed bark.

The VTA approach bridges tree biology and tree mechanics, considering not only the ability of the tree to repair itself as a way to determine potential internal defects, but also proposing the use of diagnostic instruments to probe the tree for internal defects, further probing the extent of internal defects, and eventually assessing tree risk through strength analysis of the remaining wood. This visual inspection combined with the probing tool assessment method improves the scientific nature of the assessment, and its strength analysis provides the scientific basis for computer simulation and material mechanics for tree hazard assessment.

### 3.3.2. Tree Probing Technology

The ISA classifies assessments into three levels (ANSI A300 Part 9-Tree Risk Assessment): limited visual assessment, basic assessment, and advanced assessment. The risk level of most trees can be determined from the first two levels of assessment. However, there are some trees with differences in crown morphology, internal trunk defects, and root distribution that cannot be accurately determined from external performance alone, and therefore require advanced assessments with diagnostic tools and probing instruments for in-depth inspection. This information includes the wind resistance of the canopy, wood material properties, load-bearing capacity, internal trunk air rot, and root system distribution. These inspections often require measuring instruments and testing techniques and are divided into three parts: crown, trunk, and root. Among them, the crown reflects the growth vigor and productivity of the tree. The structure of the crown is one of the main influencing factors of the vibration characteristics of the tree. The structure of the branches and twig determines the shape of the crown. The trunk, the part that connects the crown and the roots, has an important supporting role, and the decay and hollowing of the trunk is an important cause of tree decay. As a vital organ for transporting water and nutrients and anchoring the plant, the distribution of the root system makes its interaction with the soil a determining factor in maintaining the excellent state of the tree, thereby having a more significant impact on the wind response of the tree.

Controlled pulling tests are mainly used to study the wind-resistant deformation ability of tree crowns [94]. In order to simulate the bending and twisting deformation of a deflected canopy tree under wind load, a rope was attached to the center of the main trunk, and a horizontal tension was applied to produce a combined bending and twisting deformation of the trunk. A force transducer recorded the tensile force (Figure 2a). In contrast, the dynamic tree method, which uses barometers, tilt meters, and elastometers,

provides tilt measurements under actual wind load conditions, providing realistic measurements compared to traditional controlled pulling tests and eliminating the need for human monitoring of the test process. The action of wind loads on trees is a dynamic process, so it is necessary to study the wind vibration characteristics of trees under wind loads. Kolbe (2022) [95] compared the response of trees under natural wind conditions and verified the applicability of using non-destructive pulling tests to quantify wind loads. Due to the complex structure of tree crowns and branches, researchers usually reduce tree trunks and branches to certain mechanical structures [96–100]. Numerical simulations are performed by building abstract mathematical models to investigate trees' wind vibration characteristics and fall resistance under wind loads [101]. The wind vibration test chose an artificial wind source to simulate the motion characteristics of the actual wind, and acceleration sensors and data acquisition instruments collected the vibration data of the trees (Figure 2b) [88,102,103]. In order to achieve a more accurate acquisition of tree structure and physiological indicators, LiDAR technology is applied to tree information acquisition [104]. They combine computer vision and imaging algorithms, which can accurately invert tree data such as branches and trunks from scanned point cloud data, providing a basis for building three-dimensional models of trees. Jackson (2018) [105] applied dynamic modeling and analysis of complex broadleaf trees using a combination of terrestrial LiDAR and finite element analysis. Giachetti (2022) [106], in combination with finite element analysis, made it possible to identify the intrinsic frequency of trees under trees really called possible, providing a new direction for dynamic detection of trees. As shown in Figure 2b, the canopy model built by Xiao Huang et al. [107] based on the point cloud data of trees can simulate the wind resistance performance of the canopy under a wind field.

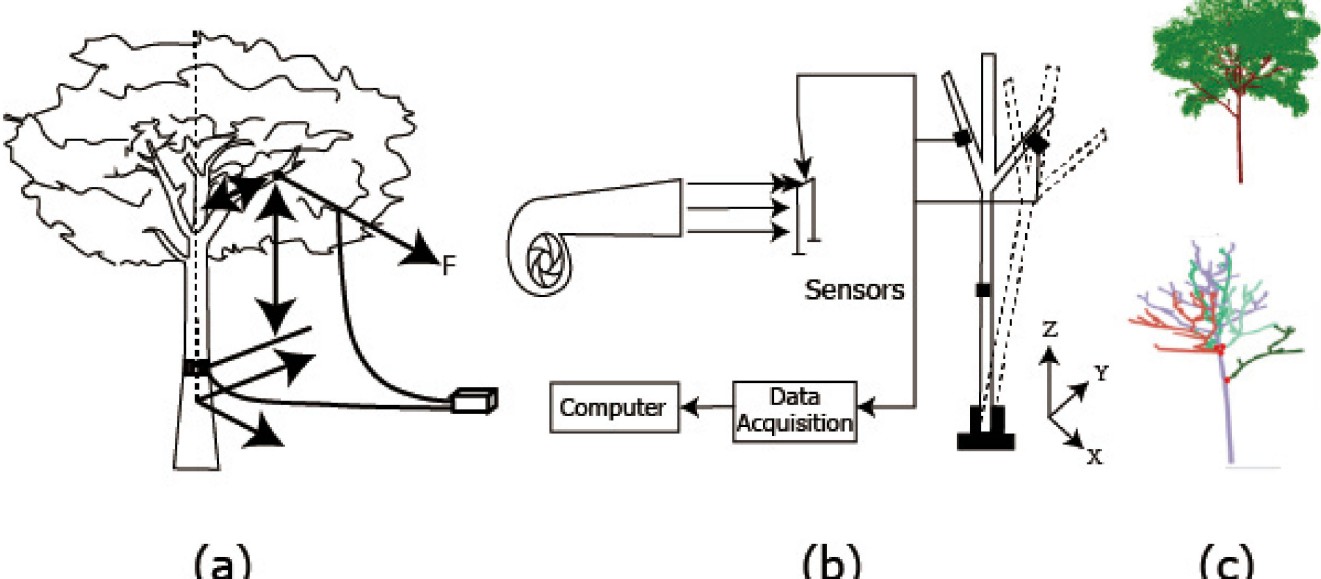

**Figure 2.** Methods of studying the wind resistance of tree canopy. (**a**) Tensile simulation; (**b**) wind vibration experiment; (**c**) LiDAR scanning.

The trunk detection is mainly to determine the internal hollow rot of the tree trunk, and the traditional trunk detection methods can cause some degree of damage to the tree. In addition to simple diagnostic tapping methods, holes are drilled in the wood to detect decay and cavities. As Table 1 shows, the traditional methods of detecting empty rot can cause varying degrees of damage to the xylem of the tree and may cause the spread of decay.

**Table 1.** Traditional methods of detecting decay and cavities [19,108–113].

| Method | Tool | Judgment | Advantages and Disadvantages |
|---|---|---|---|
| Percussion diagnosis method | Wooden hammer or rubber hammer | Change in sound of knocking trees | Easy to operate, simple tools, no damage to trees, but highly subjective |
| Drilling method | Increment borer | Observe the color of the extracted wood core to determine the decay and degree | Generally used for moderate and heavy corrosion detection, but the obtained wood cores vary greatly and are susceptible to new damage caused by the spread of decay |
| | Fractometer | Measurement of wood strength properties to determine the degree of decay | Growth cone sampling is required to quickly obtain wood properties of trees |
| | Boroscope | Drill holes in the tree trunk and use a small camera to observe the interior | Allows visual confirmation from the inside, with the same defects as the growth cone |
| Resistance measurement method | Resistograph | Insert the drill bit into the tree, measure and record the drilling resistance | Fast and easy to perform and interpret graphs, but only detects severe decay and cavities, requiring a control group |
| Resistance method | Shigometer | The xylem is drilled and a probe with pulsed current is added to determine the change in resistance | Detects early-stage decay |

Given the low accuracy of traditional methods and their tendency to cause hard-to-recover damage to trees, non-destructive testing techniques have been widely adopted because of their advantages of non-destructive and rapid detection by taking advantage of the material properties of trees and the differences between internal defects and healthy wood. NDT techniques are mostly non-invasive, causing only penetrating bark or very small wounds that do not damage the tree [114], or even non-contact inspection. NDT techniques used to detect internal defects in tree trunks are stress wave, ground-penetrating radar, resistance method, etc. To detect the presence of internal defects, a two-dimensional map of defects inside the tree can be obtained by means of laminar imaging.

Stress-wave laminar imaging is a technique that generates images of the internal structure by recording the difference in the propagation velocity of stress waves inside the tree. As shown in Figure 3a, by using multiple sensors (typically 8 to 32) to measure stress wave transmission times in multiple directions, decayed or degraded wood reduces the propagation velocity of the stress waves. The perceived velocity is then inverted to generate a two-dimensional tomographic image [115]. Commercially available acoustic tomography tools (e.g., ArborSonic 3D acoustic tomography, PiCUS Sonic tomography, and ARBOTOM®) are already available for urban tree measurements and 2D tomography image acquisition. The inversion algorithm, acoustic frequency, and number of sensors all affect the resolution of stress-wave laminar-based imaging [116]. Wei et al. (2021) [117] established a defect detection method for larch (*Larix gmelinii*) propagation law of twelve-directional stack imaging (TDSI) steps system to obtain good quantitative defect detection results.

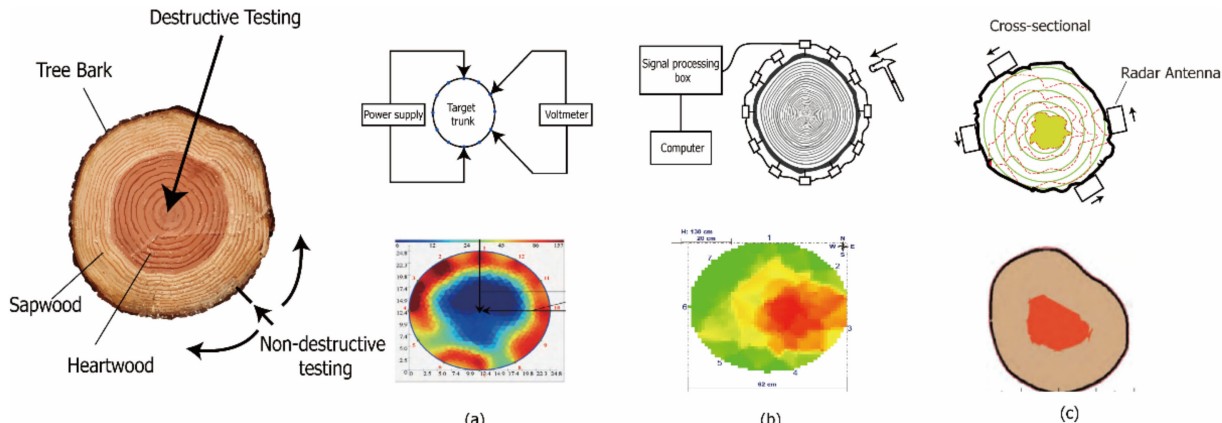

**Figure 3.** Defect detection of trunk. (**a**) Electrical impedance tomography (source: [118,119]); (**b**) stress-wave laminar imaging; (**c**) radar wave tomography (source: [120]).

Electrical impedance tomography (EIT) provides resistance images of investigated wood by measuring its electrical conductivity [121], as shown in Figure 3b. Different types of wood and changes caused by decay induce variations in resistivity values. A 2D (two-dimensional) image of resistance detection is created by calculating the discrete network on the cross-section through the inversion algorithm, the resistance value of each point after gridding, and the different values assigned to different pixels after digital image processing [115,122,123]. EIT pinpoints the location of early decay, even when there is no significant change in density.

Radar wave tomography utilizes ground-penetrating radar to emit electromagnetic waves and determines the distribution of the medium by studying the reflected waves of electromagnetic waves passing through the partitioned interfaces of different media [117]. As shown in Figure 3c, the detection of defects inside the trunk is scanned several times along the tangential axis of the trunk cross-section, which produces differences in the echo signal when defects are present inside the trunk. The echo signal is pre-processed to the elimination of spurious and noise and signal gain, offset imaging calibrates the actual position of the echo signal and then generates the internal defect map of the trunk by coordinate conversion. Since the outer contour of the trunk is mainly irregular, obtaining the outer contour information is necessary to correct defects. Each stage of the different improvement methods can improve the accuracy of the generated images. Radar wave tomography is able to accurately diagnose early and late wood decay and cavities in tree trunks. Radar wave tomography is a non-destructive method that can effectively estimate the volume of the cavity inside the trunk.

Nuclear magnetic resonance (NMR) and microwave scanning techniques can be applied equally to the detection of cavity decay in trees. NMR processes induction coils and eddy currents to reflect the passive electromagnetic properties of materials [113] (PEP), such as electrical conductivity, magnetic permeability, and dielectric constant. Detecting internal defects by analyzing PEP distribution enables effective identification of the early stages of fungal decay inside wood before decay is observed externally. MRI is a non-destructive and non-contact inspection technique, but it is not yet mature and has high equipment costs [124]. Microwave scanning techniques detect microwave absorption and scattering by wood tissues. Decay, cavities, and other internal defects of trees can be detected by estimating the attenuation, out-of-phase, and polarization of microwaves [125]. However, the imaging results can be affected by the complex material properties of wood.

The detection of root location is an essential basis for judging the stability of trees, and the root has a complex structure divided into woody and non-woody roots [126]. Among them, woody roots are roots that have undergone secondary growth and have a more rigid structure to stabilize the tree in an upright position. In contrast, non-woody roots have what are called fine roots that are only responsible for absorbing water and nutrients and

are less than 2 mm in diameter. Most of the lossy root detection methods require excavation to observe the root system, which is costly, time-consuming, and damaging to the tree. The assessment of trees is mostly performed using non-destructive testing methods that do not harm or damage them and are easily repeatable, which means that long-term investigation and monitoring of trees can be achieved [127].

Ground-penetrating radar can detect most of the coarse roots based on the difference between the root and soil dielectric constants. [128]. Moving along a square or circular trajectory and recording the reflected signals generated by electromagnetic pulses, it generates a time-depth profile of the recorded subsurface, which is a two-dimensional representation of the subsurface (often called B-scan) [129]. The echo biplane represents the location information of the buried subsurface (as shown in Figure 4), therefore the B-scan plot can visualize the subsurface tree roots. The detection resolution of a ground-penetrating radar depends on the antenna frequency, the electromagnetic properties of the medium, and the penetration depth [130].

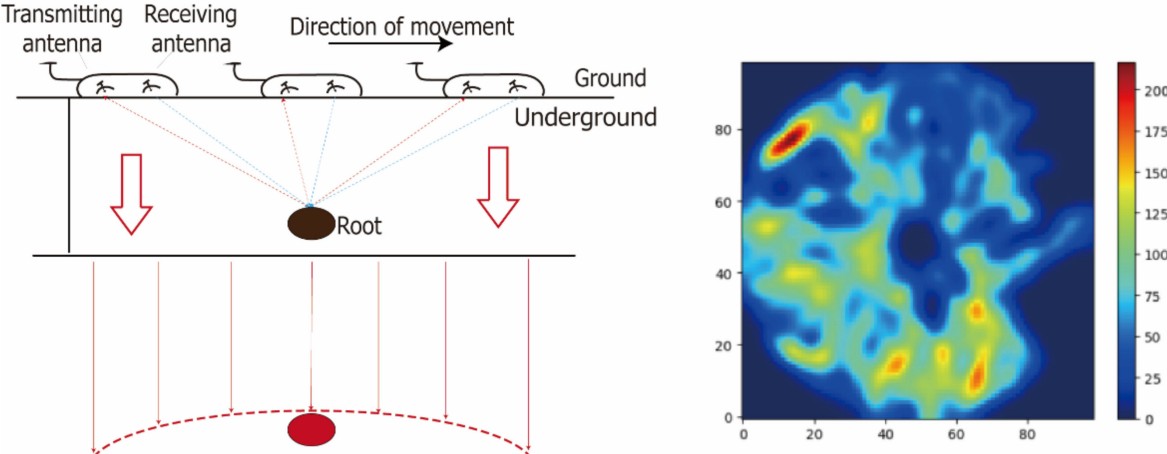

**Figure 4.** Ground-penetrating radar imaging principle and scanning images.

Ground-penetrating radar scans are subject to clutter interference from debris in the soil, so post-processing of the radar wave data is required prior to interpretation. A standard ground-penetrating radar data post-processing scheme should include radar map normalization, noise reduction, signal amplification, offset, and Hilbert transform. At the same time, for the detection of underground roots into the image of the hyperbolic extraction and the calculation of the related parameters, then map, tree roots have made new progress in recent years, making it possible to set up the roots of the 3D model [127]. Root diameter is an important parameter reflecting the geometric characteristics of the root system, and the estimation of root direct and orientation is the focus of ground-penetrating radar research. Zhu et al. [131] and Yeung et al. [132] developed a linear estimation model of root diameter using parameters such as time interval, amplitude area, etc. Liang et al. [133] proposed a root diameter estimation method based on BP neural network. Based on the estimation of root parameters for 3D reconstruction of tree roots, Fan et al. [134] developed an algorithm for automatic reconstruction of tree roots implemented using topological and geometric relationships.

Electrical imaging maps the spatial distribution of tree roots by making multiple electrical measurements in an electrode array, depending on the electrical properties (e.g., conductivity, resistivity) that differ between the soil and the roots. Electrical resistivity tomography is a non-destructive method for detecting tree roots and is widely used to study the interaction between the water cycle in the soil and the plant root system [135,136]. Amato et al. (2008) [137] found a significant relationship between root mass density and resistivity values, which can be used for tree root detection. Giambastiani (2022) [138] compared ERT results for different soil pointing and moisture levels and found

that the root system affects the resistivity of the surrounding soil, providing a prerequisite for non-destructive root detection. In most cases, a "rough" mapping of root structures, distributions, extensions, or structures can be achieved, but individual root measurements are still difficult because of the limited spatial resolution of the method. Other non-destructive testing methods, such as the rhizotrons and minirhizotrons method, resistance method, acoustic detection method, X-ray method, etc. can be applied to the root system's detection. The rhizotrons and minirhizotrons method involves inserting transparent plastic tubes into the soil or using glass plates and using camera equipment to acquire images of tree roots, allowing observation of the development and growth of the tree root system. Mohamed (2017) [139] compared the different methods of image acquisition and pointed out that inexpensive scanning and automated methods can correctly measure root elongation and length. Bucur (2015) [140] proposed a method to detect the location of tree roots using sound by exploiting the difference in the propagation speed of sound in soil and tree roots. Similar to stress-wave detection of tree trunks, tapping the transmitter on the tree trunk and receiving the signal from the ground near the trunk can detect some roots up to 0.5 m in depth using the difference in reception time. X-ray computed tomography uses X-rays to obtain two-dimensional cross-sectional images of scanned objects from which three-dimensional reconstructions can be made [141]. In a study of tree roots, Pierret (1999) [142] used CT on chestnut (*Aesculus hippocastanum*) and maple (*Acer pseudoplatanus*) trees for the distribution of tree roots under undisturbed soils and proposed a series of methods from field sampling to spatial analysis. Kaestner (2006) [143] reconstructed the high-fidelity fine roots of alders (*Alnus incana* (L.) Moench) using an improved algorithm based on diffusion filter to enhance the contrast between roots and sand. Jassogne (2008) [144] used RootViz (developed by Davidson, www.rootviz3d.org, accessed on 29 August 2022) to segment saltbush (*Atriplex hortensis*), lucerne (*Medicago sativa*) and canola (*Brassica napus* L.) in 3D root structure, which demonstrates the potential of medical CT in visualizing large roots. In a recent study of tree roots, Park (2022) used CT to examine the structure of roots that had been severed and then repaired for growth, demonstrating the potential of CT to be used in forest pathology. Table 2 shows the primary non-destructive root system detection.

**Table 2.** Advantages and disadvantages of main non-destructive root system testing methods [126].

| Method | Advantages | Disadvantages |
|---|---|---|
| Rhizotrons and minirhizotrons | High resolution imaging and repeatable measurements | May affect the root growth and only a part of the roots can be observed, high cost and limited installation |
| Ground penetrating radar | Accurate diagnosis of early and late wood decay and trunk cavities, and calculation of cavity volume | Detection of wood layers requires high-resolution frequency domain methods |
| Electrical resistivity tomography | Easy data collection and repeatable measurements 1D, 2D, and 3D measurement capabilities | Systematic errors due to poor electrode contact exist Longer measurement time Difficult to discriminate the effect of roots from the background noise of low root biomass |
| Acoustic detection | Detectable thick roots | No detection of small roots (<4 cm diameter) Shallow detection depth (<50 cm) High sensitivity to water content Difficult to distinguish roots from other buried materials |
| X-ray computed tomography | High resolution imaging Easy repeatable measurements Fine root detection | Difficult to distinguish between roots and other materials High dependence on soil-related factors (i.e., soil type, soil moisture content, presence of organic matter or aerated pore space, root moisture status) Overestimation of root diameter and short root length |

## 4. Discussions

The development of tree risk assessment has been accompanied by continuous improvement in the in-depth study of trees. Systematic tree risk assessment summarizes the factors affecting tree risk from the characteristics exhibited by failed trees at the accident site and selects a reasonable approach to determine the overall risk value. This approach effectively predicts tree risk from a simple visual perspective, but reliance on the assessor's professionalism and subjective judgment is unavoidable. Therefore, when developing and using assessment methods, it is important to objectively consider the conditions of application of the method and to determine the overall risk in a way that truly reflects the conditions of application of each factor.

In order to provide a more scientific basis, research in tree mechanics has introduced some judgmental features for risk assessment that can reflect the tree's carrying capacity. The structure of the tree, damage, and adaptive growth characteristics can all be used to assess the possible risk of the tree. When introducing characteristics related to tree mechanics, measurement tools are often required to assist in obtaining them, and it is essential to consider that the characteristics are easily accessible and relatively accurate.

Generally, the canopy is subject to the action of branches on wind loads, the loss of strength due to defects within the trunk, and the distribution of roots on the anchoring effect of the tree and its contribution to stability, all of which cannot be judged by simple observation alone. Therefore, some probing tools and theoretical methods are needed to achieve this. With the development of technology, many probing techniques using different physical responses of trees (e.g., resistivity, different wave propagation velocities, etc.) have been applied to tree detection, and have evolved from causing damage to trees to non-destructive detection, allowing more and more efficient and convenient consideration of tree characteristics. Taking into account the complex structure of trees, theoretical methods have evolved from simple ideas to add considerations that allow the methods to be closer to the actual conditions of the trees. There is a trend that many probing tools can model trees using finite element methods to put numerous factors into calculations, such as the response of complex branch structures to dynamic wind loads, the calculation of strength loss of trunks using actual cross-sectional shapes where empty rot exists internally, and the calculation of anchorage capacity of underground trunk distributions. When using NDT tools, the resolution of the measurement, the cost, and the measurement time need to be considered. The location of trees may present obstacles to NDT, such as buildings and underground pipes, so using the appropriate inspection tool for each environment is necessary.

The special spatial distribution of large urban development, global warming, and environmental pollution pose challenges to the growing conditions of trees. Trees are inevitably at risk to varying degrees in order to grow on their own and to cope with the potential damage caused by the complex external environment. Therefore, a scientifically sound risk assessment and risk management of tree impacts is more than indispensable. At the same time, we should also plan the growing space of trees rationally, as well as protect the environment.

## 5. Conclusions

Current research on tree risk is mainly based on the causes of accidents to speculate on specific defects and structural impact principles. Systematic risk assessment is a reasonable and economical way to initially determine tree risk, but data acquisition for assessment metrics is still predominantly subjective. Adding evaluation tools for objectivity, combining modern technologies such as computer vision for accurate data collection, and using machine learning to determine metric weights and develop a professional tree risk assessment website or application that makes the initial risk assessment a simple, quick and reusable assessment method are all prospective research directions for determining tree risk.

Visualized tree risk assessment methods allow for further quantification and comprehensive objective evaluation of the complex factors affecting tree risk. Existing assessment

methods include research and exploration of internal tree defects and root distribution. The next can be used to analyze tree strength or stability with the help of visual assessment results combined with mechanics, by analyzing the loss of tree strength due to wood decay or the presence of tree defects. This analysis requires the development of a complete model of the overall strength and stability of the tree. Refined modeling using both mechanical and mechanical methods is used to analyze the change in strength of trees under the influence of loads such as wind, erosion, and canopy effects. Numerical simulations can be performed using finite element methods to predict the risk of individual cases based on different defects and structural characteristics of different trees, which would be a potential research direction for applying the test results to risk assessment.

Given the number and distribution characteristics of urban trees, different assessment and management efforts should be applied to different trees. A systematic risk assessment of all trees should be conducted to distinguish the risk level of different trees before conducting a fine-scale assessment. Trees with higher risk levels should be finely assessed. Therefore, the establishment of a complete tree risk assessment system to realize the graded and hierarchical management of different trees will probably also be a research hotspot in the modern management of urban trees.

**Author Contributions:** Conceptualization, H.L.; methodology, H.L.; formal analysis, H.L. and X.Z.; investigation, H.L.; writing—original draft preparation, H.L.; writing—review and editing, H.L.; visualization, H.L., Z.L. and X.T.; supervision, J.W.; project administration, J.W. All authors have read and agreed to the published version of the manuscript.

**Funding:** This study was supported by the National Natural Science Foundation of China (Grant No. 32071679) and the Beijing Municipal Natural Science Foundation (Grant No.6202023).

**Conflicts of Interest:** The authors declare no conflict of interest.

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
