# Peer review of "A Review of Research on Tree Risk Assessment Methods"

_forests, doi:10.3390/f13101556_

Round 1

Reviewer 1 Report

The article is a review of tree risk assessment methods. The authors presented an outline of selected tree risk assessment methods, assessed the advantages and disadvantages of their application. They also presented and assessed selected techniques of stump health structure assessment used in the study of tree statics and stability. The aim of the review of these studies was to determine the further direction of development of the tree risk assessment.

The authors compared the definition of risk and threat. They discussed the history of visual tree assessment methods, focusing mainly on American and Chinese research. Although German (VTA), English (QRTA Ellisson) and Polish (Bruchwald) methods have appeared. In addition, they discussed the operation of techniques and tools for assessing the health of trees, both invasive (increment borer, fractometr, resistograph, boroscope, shigometr) and non-invasive (tomography, GPR, etc.).

Among the technical methods of statics assessment, the authors characterized the test techniques, including only Tensile tests.

Although the article can be assessed positively as a whole, it contains several shortcomings and inaccuracies that should be corrected.

It is worth emphasizing that the first methods of tree assessment are derived from forest stock surveying of standing trees. The first methods of tree static/stability assessment are based on a checklist  that indicate a weakening of the mechanical functions of trees. This list was similar to the classification of wood defects used in forestry.

A clear distinction should be made between the presence of statics assessment methods (VTA, IBA, SIA, Biostatische Baumkontrolle - and therefore mainly German) and risk assessment methods (originally American - BMP, ISA, later European (Ellisson, WID-method etc.).

It should be emphasized the difference between the methods of assessing urban trees (eg VTA) and forest trees (Bruchwald

The authors rightly argue that any tree can be a hazard, but also that the risk of a fatal accident is very low, as has been studied in England, for example. Therefore, visual methods simplify the assessment of the likelihood of a tree falling by introducing parameters such as exposure to the threat, severity of undesirable effects or the possibility of minimizing the risk. This makes sense given the low severity of risk and the large benefits of trees (ecosystem services).

The authors paid little attention to visual assessment methods based on observational data and on the principles of biomechanics (e.g. SIA, Wessolli method). They did not cover a number of other test methods, including stress tests, which are frequently used in Europe and which have many limitations and may represent a misjudgment.

The article lacks a critical evaluation of the methods, both visual (e.g. one critical threshold for the slenderness ratio, the criticized t / R ratio) and a clear discussion of the limitations in the use of technical methods (multi-stem trees, reaction wood, wet heartwood, tree defense mechanisms, etc. .). Some techniques (e.g. tensil tests, dyna-tree, dyna-root) do not have a developed reference scale for the results obtained.

In conclusion, there is no assessment of the limitations in the use of existing methods and a clear definition of the needs in their improvement, or perhaps simplification.

The article can be accepted for publication after corrections of comments

Reviewer 2 Report

The review aims to retrace the path regarding risk assessment. it is roughly articulated without following the logic of the theme well (hazard -> vulnerability -> risk). The methods used and the tools adopted to evaluate a tree are reported.

in a superficial way, as there is a lack of many elements of which the literature is full. Some issues are addressed based on publications from 10 years ago, without considering what was published later. The discussions do not leave much in terms of reflections or criticisms in relation to the above, but they just summarize everything.

Some concepts are poorly reported compared to citations, or commonly used names and concepts are not reported. Themes that have made history in the sector, such as Mattheck's constant streess hypothesis.

From my point of view, we need a greater bibliographic study, a re-organization of the review and discussions that leave something to the reader.

52 - This sentence shows some gaps in terms of the definition of risk, hazard and target. The risk assessment already includes the hazard of the tree and the vulnerability of the targets. So it is quite a contradiction I would say that it should often be done in places where we have high vulnerability. Rather it should be said that in these places constant monitoring of the stability of the trees is necessary.

74 - I really did not understand this paragraph, I can only emphasize that risk and target are two very different things.

78 - The health of the tree directly influences the ability of the tree to react to loads, according to the processes called (mechanosensing, thigmomorphogenetic) - in particular in this review these concepts related to the works of Moulia et al., Gardiner et al. ... etc.

217 - !

219 - define the symbols of the equation

241 - the same of Eq 1

289 - this whole part related to the root system is unclear and non-linear. It seems that the author has taken some concepts from the texts cited without following a logical thread.

Some notes: row 295 - we don't use "soil" but "root-plate" - what moves is the whole complex consisting of soil and roots.

299 - this phrase "Root-soil plates play an important role in tree stabilization" doesn't make any sense. The root-soil plates are the seat of the anchoring, they have the function of anchoring ... saying "play an important role" indicates that they affect the stability as if they were an external factor or a specific element.

295 - root plate, not "soil"

299 - ????

305 - VIRTUAL???? ...VTA = Visual

308 - not only....snow, manufact impact, rain,...

310 - it's called "Mattheck's constant stress hypothesis", why don't you report these concepts in the review?

332 - in the VTA method we do not speak of risk but of hazard. As the context is not taken into account. The VTA done inside a wood or a square in the city center gives identical results, while the risk is widely different.

345 !!!

365 - throughout the literature it is referred to as a controlled pulling test, no "tensile"

459 - this sentence seems a bit too optimistic to me. the author quoted in the conclusions reports ". However, the detection and quantification of coarse roots using GPR is still in its infancy and not all roots or soil conditions are suited for this technology". After this work, we find in the literature an interesting work by Tardio using the GPR, but even here we do not find such a solid statement.

Other recent publications of other authors regarding NDT, such as Giambastiani, Zenone, Morelli, Amato, Cassiani, are not taken into consideration. In general, this part shows gaps in bibliographic references, for example table 3 summarizes the details of the techniques illustrated, without mentioning the origin of the information (essential for a review).

Many works that study biomechanics are also not taken into consideration, in particular in recent years we have seen very developed dynamic analysis (using accelerometers and other techniques, just mentioned). Important works are by Abbas, James, Giachetti, Sterken, Sani, Fourcaud, Telewski ...

484 - the discussions of a review should take up everything previously reported and make a contribution on the development of the theme. Here, besides a summary, there is no food for thought. ...What are the current needs, how to decline everything in the phenomena we are witnessing today (such as climate change, the development of megacities, urban pollution, etc).

Reviewer 3 Report

See comments in attachment

Round 2

Reviewer 2 Report

The authors carried out a thorough review, following the indications of the reviewers, in a timely manner, responding to comments and accepting some observations.

The work appears more complete, updated to the state of the art and perfected in terminology and concepts.

Reviewer 3 Report

This article is improved